# PersonalQ: Select, Quantize, and Serve Personalized Diffusion Models for Efficient Inference

## Abstract

Personalized text-to-image generation enables users to create custom AI models that generate their unique concepts—specific objects or artistic styles—achieving unprecedented creative control. However, deploying a large repository of personalized checkpoints faces two critical challenges: (1) ambiguous user prompts make it difficult to match the intended checkpoint in large repositories, and (2) standard post-training quantization methods degrade personalized diffusion checkpoints' image quality. We analyze the importance of reasoning over checkpoint metadata and clarifying user prompts for intent-aligned checkpoint selection. Additionally, we find that trigger tokens for personalized diffusion play a crucial role in quantization. To address the challenges, we propose PersonalQ, a unified system with two components: *Check-in* analyzes checkpoint repositories and clarifies user intent for intent-aligned selection, and *TAQ* (Trigger-Aware Quantization), which protects the trigger-token-related representation to deliver high-quality inference from the chosen checkpoint under quantization. On our REPO-PROMPTS benchmark, PersonalQ achieves an 89% checkpoint-selection preference win rate and a 4.42/5 intent score. Across benchmarks, *TAQ* reduces inference memory by up to 75% while maintaining strong text-image alignment (CLIP score 0.297 vs. 0.315 at full precision) and image fidelity (FID 11.03 at W8A8 vs. 10.96 at full precision), enabling scalable deployment of personalized models without compromising quality.

## 1 Introduction

Diffusion-based (Sohl-Dickstein et al., 2015) text-to-image models (Rombach et al., 2022; Saharia et al., 2022; Podell et al., 2023; Li et al., 2023b) can be personalized through fine-tuning methods such as DreamBooth (Ruiz et al., 2023) and LoRA (Hu et al., 2022). These methods create personalized checkpoints—model weights that have learned to generate specific concepts when prompted with special trigger tokens. For instance, a checkpoint fine-tuned on images of a specific teddy bear can generate that exact bear when prompted with trigger tokens `<bear-v4>`, while one trained on a unique painting style could respond to trigger tokens `<painting>`. As users accumulate personalized models, they build large repositories containing dozens or hundreds of checkpoints—multiple versions of the same concept, different artistic styles, various training timestamps. To generate images, users must select the appropriate checkpoint from their repository using natural language descriptions like "Bear on forest grass using the April-trained version," which may reference training dates, visual characteristics, or other metadata. However, current model selection relies on manual browsing or other retrieval methods (Luo et al., 2024) that struggle to match these nuanced prompts to the correct checkpoint (Figure 1 a). This scenario raises an important question (**Q1**): How can we effectively match user prompts to their preferred checkpoints within a large repository?

Assuming the user selects the preferred checkpoint, serving these personalized models at scale introduces severe GPU memory constraints. When serving personalized models, each inference request must load the corresponding checkpoint into GPU memory for generation. A server handling multiple concurrent users—each requesting different personalized models—quickly exhausts GPU memory. For example, serving just 10 users simultaneously with different personalized models (each requiring 4GB) demands 40GB of GPU memory, exceeding most consumer GPUs. The problem worsens on edge devices with limited memory. A natural solution is quantization, which reduces model size by using lower numerical precision. However, personalized models break under standard quantization approaches. Even recent quantization methods (Li et al., 2023a; Huang et al., 2024; Liu et al., 2024; Ryu et al., 2025) fail to preserve personalized content quality (Figure 1 b). This leads to our second question (**Q2**): How can we reduce GPU memory requirements for serving personalized models while maintaining high image quality and text-image alignment?

We introduce PersonalQ, a unified framework that addresses both critical questions for serving quantized personalized model repositories: (**Q1**) intent-misaligned checkpoint selection and (**Q2**) Quantization methods designed for general diffusion models can degrade image generation quality in personalized diffusion models. To address Q1, we recognize

Figure 1: **Problems**: (a) Existing checkpoint selection methods misdirect user queries to wrong checkpoints, producing off-intent outputs. (b) Standard post-training quantization methods degrade personalized models. **Solutions**: (c) An intent-clarification agent retrieves and filters to select the appropriate full-precision checkpoint for the query. (d) We then quantize that checkpoint while protecting trigger-token–related features, enabling efficient inference with personalization preserved and reduced memory footprint.

that simple retrieval methods fail at checkpoint selection—they cannot effectively filter structured metadata (timestamps, style tags) or interpret complex user requirements. PersonalQ solves this with the *Check-in*, which performs metadata-aware reasoning to accurately match user instructions with appropriate checkpoints (Figure 1 c). *Check-in* achieves this by: (1) analyzing both visual descriptions and structured metadata (timestamps, style tags) to understand user intent, and (2) detecting ambiguous instructions that require optional user clarification. This approach ensures reproducible and accurate checkpoint selection.

For **Q2**, we first observe that trigger tokens in prompts (special placeholders that activate learned concepts; e.g., <bear-v4> for 'bear') are particularly sensitive to quantization within the cross-attention blocks (Figure 2). Our analysis reveals that existing methods disregard this characteristic, resulting in significantly larger quantization errors compared to full precision (FP32) outputs and degraded text-image alignment. Based on this finding, PersonalQ introduces Trigger-Aware Quantization (TAQ), which treats trigger tokens separately from general tokens (Figure 1 d). TAQ partitions cross-attention key/value and hidden-state tensors by token semantics, applies targeted quantization only to non-personalized components, and concatenates them. This approach enables us to achieve high text-image alignment without compromising image quality while reducing memory and latency.

We introduce REPO-PROMPTS, a dataset that blends content descriptions with metadata cues to evaluate checkpoint selection. We evaluate across a repository of 1,000 personalized checkpoints spanning objects, characters, and styles. On REPO-PROMPTS, *Check-in* achieves an 89.1% LLM-judge win rate, a 4.42/5 intent score, and resolves 89.3% of ambiguous prompts. *TAQ* maintains high fidelity at practical bit-widths: W8A8 attains FID 11.03 and CLIP 0.297; W4A8 attains FID 13.74 and CLIP 0.292, close to full precision (FID 10.96, CLIP 0.315). These settings reduce parameter storage by 4–8× and bit-operations by 16–32×, and cut serving-time GPU memory by up to 75% relative to FP16, while maintaining personalization quality.

To the best of our knowledge, we are the first to achieve intent-aligned and memory-efficient serving of personalized diffusion models under quantization settings.

In summary, this paper makes the following contributions:

- We introduce *Check-in* to reliably selecting the user's intended personalized checkpoint. Given a natural-language request, Check-in jointly reasons over repository metadata—such as timestamps (to capture recency and version

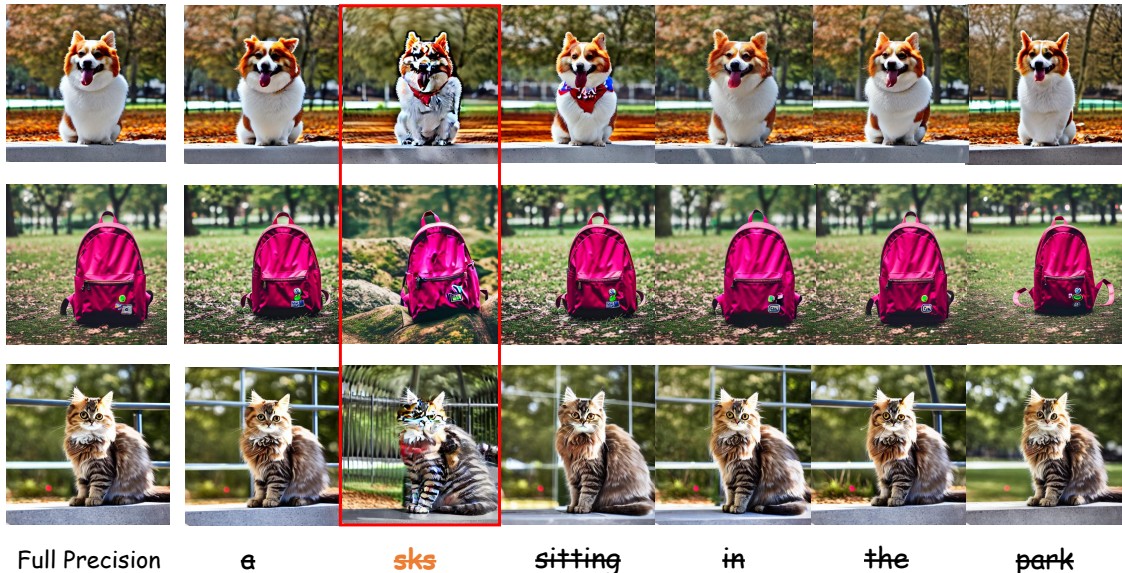

(a) Token-wise sensitivity. Only the indicated token is quantized; all others stay full precision. Quantizing the personalized trigger `<sks>` (red box) causes the largest visual degradation.

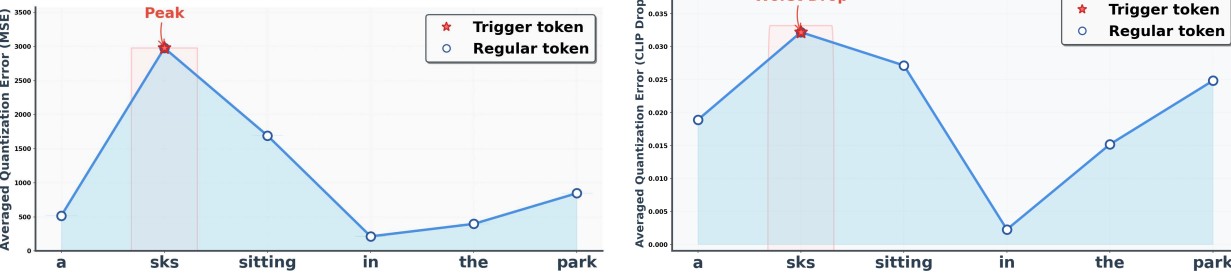

(b) Averaged MSE when quantizing each token (higher is worse). Error peaks at the trigger token.

(c) Averaged CLIP-score drop when quantizing each token (higher is worse). The trigger token yields the largest drop.

Figure 2: **Trigger token (e.g., `<sks>`) is vulnerable under quantization.** We measure per-token sensitivity to cross-attention Key/Value-row quantization by quantizing only one token at a time while leaving all others in full precision. Visual results and aggregate metrics show the trigger token is far more fragile than common words under 4-bit quantization. *Prompt used:* "a `<sks>` sitting in the park", where `<sks>` denotes the learned trigger token for different personalized checkpoints.

history) and style tags (to summarize training intent)—as well as over visual descriptors derived from example images. It detects and resolves ambiguous intent by surfacing underspecified attributes, when needed, prompting for a brief clarification. Finally, it maps generic nouns and colloquial terms to library-specific trigger tokens, ensuring the downstream model receives precise cues.

- We present *TAQ*, a post-training quantization method for personalized diffusion models. TAQ keeps 32-bit precision along trigger-token cross-attention pathways and quantizes the rest, yielding 4×–8× smaller parameter storage and 16×–64× fewer bit-operations than FP32 with minimal quality drop. On MS-COCO: FP32 achieves FID 10.96 and CLIP 0.315; under 8-bit weights and 8-bit activations (W8A8), FID 11.03 and CLIP 0.297; under 4-bit weights and 8-bit activations (W4A8), FID 13.74 and CLIP 0.292.

- We release and evaluate on a 1,000-checkpoint personalized repository and our REPO-PROMPTS benchmark for metadata-aware selection. *Check-in* attains an 89.1% LLM-judge win rate with 4.42 (scale 1–5) human appropriateness and resolves 89.3% of ambiguities, validating accurate selection alongside TAQ's favorable memory–quality trade-offs across settings.

## 2 RELATED WORKS

### 2.1 PERSONALIZED TEXT-TO-IMAGE GENERATION

Personalized generation methods teach pre-trained models to generate specific subjects (like your pet or face) by showing them a few example images. DreamBooth (Ruiz et al., 2023) achieves this by fine-tuning the entire model using a special trigger word (e.g., "a photo of `<sks>` dog") to represent your specific subject. Textual Inversion (Gal et al., 2022) only learns what the trigger word means without changing the model itself. LoRA (Hu et al., 2022) finds a middle ground—it adds small trainable modules to the model, making personalization faster and requiring less storage. While these methods successfully create personalized models, they don't solve the practical problem: when users accumulate hundreds of personalized models, how do we automatically pick the right one from ambiguous requests and serve it efficiently on limited hardware?

### 2.2 CHECKPOINT SELECTION AND MANAGEMENT

Selecting the right model from multiple versions is challenging. Existing methods like Stylus (Luo et al., 2024) match artistic styles for LoRA adapter selection, while Mix-of-Show (Gu et al., 2023) merges adapters for combined effects. However, these assume users know their exact needs and ignore practical constraints like version history and memory limits. Standard retrieval methods like RAG (Lewis et al., 2020; Zhu et al., 2024) or reranker (Ma et al., 2023) work for text but fail with model metadata and ambiguous requests (e.g., "use the latest cat version with artistic style"). While reranking approaches (Niu et al., 2024) leverage language model reasoning, our *Check-in* uniquely combines prompt understanding with checkpoint metadata analysis—including timestamps, styles, and hardware requirements—to resolve ambiguous requests within resource constraints.

### 2.3 POST-TRAINING QUANTIZATION OF DIFFUSION MODELS

Post-training quantization (PTQ) reduces model size and accelerates inference by replacing high-precision weights and activations with low-bit representations (e.g., 4 or 8 bits). Diffusion models pose unique quantization challenges as activation ranges vary significantly across denoising timesteps (Shang et al., 2023; Wang et al., 2024). Fixed quantization parameters can severely degrade quality at low bit-widths. While existing PTQ methods use timestep-adaptive strategies (Li et al., 2023a; Huang et al., 2024) or preserve outliers (Liu et al., 2024; Ryu et al., 2025), our TAQ method specifically targets personalization. It maintains high precision for trigger-token pathways while aggressively quantizing other components, preserving personalization quality under memory constraints.

## 3 METHODOLOGY

We propose a two-part system for accurate selection and efficient, fidelity-preserving inference over repositories of personalized checkpoints: *(A) Check-in* reasons over repository metadata and prompt semantics, maps generic nouns to trigger tokens, and resolves ambiguity (Sec.3.1); and *(B) TAQ*—Trigger-Aware quantization that preserves identity-critical pathways while quantizing the rest (Sec.3.2). Together, the components preserve user intent in checkpoint selection, maintain subject fidelity under quantization, and meet throughput and memory targets, enabling scalable personalized image generation. Figure 3 illustrates our pipeline.

### 3.1 *Check-in*: MANAGING AMBIGUITY IN MULTI-CHECKPOINT SELECTION

Personalized text-to-image systems maintain hundreds of checkpoints $\mathcal{C} = \{c_1, ..., c_n\}$, each requiring specific trigger tokens for activation. User requests like "Forest's fall grass with an April-trained version of bear" require both clarifying prompt semantics and selecting the optimal checkpoint $c^* \in \mathcal{C}$. While prompt clarification has been explored (Chen et al., 2025), checkpoint selection reflecting user intent remains unaddressed.

#### 3.1.1 CHECKPOINT SERIALIZATION AND METADATA

Each checkpoint $c_i$ is represented as a tuple $c_i = (T_i, S_i, D_i, M_i)$ where $T_i$ is the set of trigger tokens, $S_i$ is the set of subject types, $D_i$ is the semantic description, and $M_i$ is the temporal metadata. For example, checkpoint `bear-v4` includes trigger tokens $T_i = \{\text{<bear-v4>}\}$, subject types $S_i = \{\text{bear, teddy bear, plush toy}\}$, visual description $D_i =$ "A small plush teddy bear, tan with a cream snout and belly", and temporal metadata $M_i = \{\text{created\_at: 2025-04-11, version: 4}\}$. This enables semantic retrieval from natural language, temporal reasoning for version-specific requests, and deterministic token mapping. We extract visual characteristics through inference, then

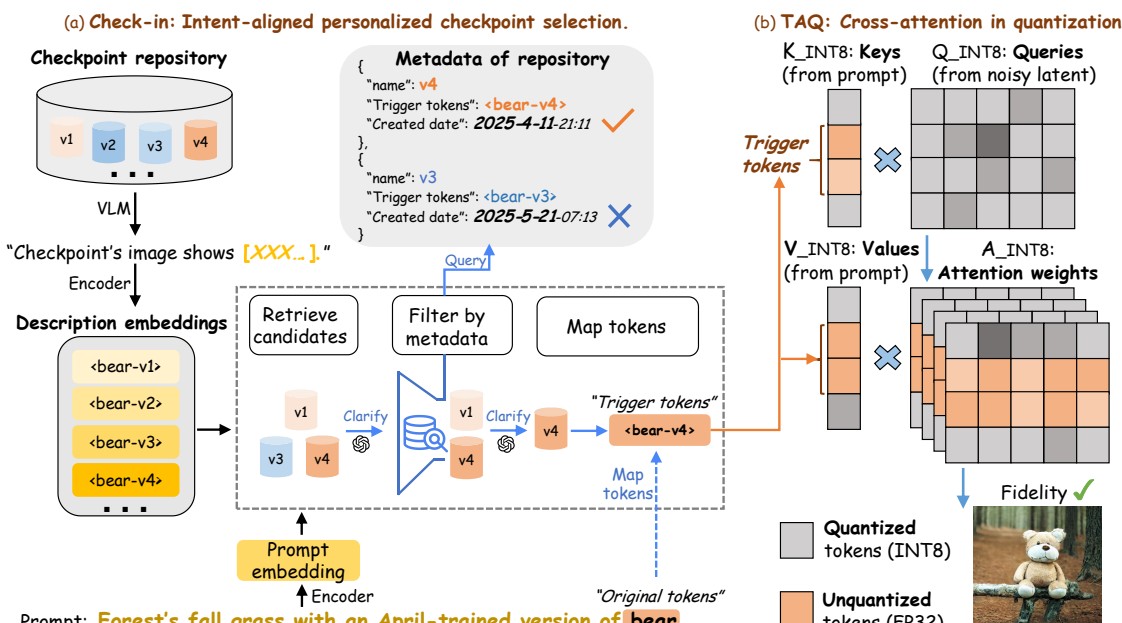

Figure 3: **PersonalQ: An inference system for quantized personalized checkpoint repository.** (a) *Check-in:* Encode the prompt and checkpoint descriptions; retrieve candidates; filter them with repository metadata (e.g., training date, trigger tokens); and map generic terms to the target trigger (e.g., bear → <bear-v4>). Output: an intent-aligned personalized checkpoint. (b) *TAQ (Trigger-Aware Quantization)* during post-training quantization of the cross-attention blocks, keep FP32 for trigger-token–related representationss and quantize the rest, preserving image fidelity while lowering memory use for efficient inference.

use Gemini 2.5 Flash (Team et al., 2023) to generate comprehensive descriptions. When a new checkpoint is created, its description $D_i$ is encoded as an embedding $e_i = E(D_i)$ using the Qwen3-Embedding-4B model (Zhang et al., 2025) as our embedding model $E$. These embeddings are then added to the pre-computed description embeddings for efficient retrieval.

### 3.1.2 SELECTION STAGE

The selection stage employs a three-step refinement process:

**Step 1: Initial Retrieval.** Given user prompt $p$ and embedding model $E$, we compute cosine similarities between $e_p = E(p)$ and the pre-computed checkpoint description embeddings $\{e_1, ..., e_n\}$. The top-$K$ checkpoints (where $K = 10$) are selected based on similarity scores $\text{sim}(p, c_i) = \frac{e_p \cdot e_i}{\|e_p\| \|e_i\|}$.

**Step 2: Metadata Reasoning and Filtering.** Deterministic rules narrow the candidate set $\mathcal{C}_K$ based on explicit constraints: For the example prompt "April-trained version of bear", the system filters $\mathcal{C}_K$ to only include checkpoints where $\text{month}(M_i.\text{created\_at}) = \text{April}$ and $S_i \cap \{\text{bear}\} \neq \emptyset$. The filtered set $\mathcal{C}_F \subseteq \mathcal{C}_K$ contains only checkpoints satisfying all applicable constraints. In our implementation, we choose Gemini 2.5 Flash (Team et al., 2023), with a 128K context window (see Appendix D for full prompt).

**Step 3: Clarification.** When $|\mathcal{C}_F| > 1$, the system extracts distinguishing features and presents an intuitive choice: "Would you prefer a photorealistic bear or a cartoon orange bear checkpoint?" This ensures users need not understand checkpoint technicalities—they simply describe their vision and answer clarifying questions when ambiguity exists.

### 3.1.3 MAPPING STAGE

Users specify content using natural language without knowing required trigger tokens. The mapping stage systematically substitutes generic nouns with corresponding triggers. For each generic noun $w$ in prompt $p$ matching the selected checkpoint's subject types $S_{c^*}$, we apply the transformation $p' = \text{replace}(p, w, t_{c^*})$ where $t_{c^*} \in T_{c^*}$ is the canonical trigger token. For the prompt "April-trained version of bear," we map "bear" to the selected checkpoint's canonical trigger $t_{c^*} = $ <bear-v4>. The resulting prompt is "April-trained version of <bear-v4>," which we then pass to the

downstream quantized inference pipeline.For the prompt "April-trained version of bear," we map "bear" to the selected checkpoint's canonical trigger $t_{c^\star} = $ `<bear-v4>`. The resulting prompt is "April-trained version of `<bear-v4>`," which we then pass to the downstream quantized inference pipeline. We use Gemini 2.5 flash (Team et al., 2023) to perform this mapping.

### 3.2 TRIGGER-AWARE QUANTIZATION: ADDRESSING THE CHALLENGE

**The need for quantization in diffusion models.** Deploying diffusion models on resource-constrained devices requires reducing their memory footprint through quantization—replacing high-precision weights and activations with low-bit representations. There are two common quantizers: uniform and logarithmic quantizers. In uniform quantization, a full-precision value $x$ is mapped to a $b$-bit integer $x_q$ using scale factor $s > 0$ and zero-point $z$. Logarithmic quantization captures distributions with large dynamic ranges by quantizing in the logarithmic domain. The mappings and reconstructions for these methods are:

$$\text{Uniform:} \quad x_q = \text{clip}\big(\text{round}(x/s) + z, \, 0, \, 2^b - 1\big), \qquad \hat{x} = s\,(x_q - z),$$

$$\text{Logarithmic:} \quad x_q = \text{clip}\left(\text{round}\left(\frac{\log(|x| + \epsilon)}{s}\right) + z, \, 0, \, 2^b - 1\right), \qquad \hat{x} = \text{sign}(x)\left(e^{\,s(x_q - z)} - \epsilon\right).$$

**Why personalized diffusion models are different.** Personalized checkpoints learn to associate specific trigger tokens with the target concept. Consider a user prompt $\tau = (w_1, \dots, w_T)$ that contains a personalized trigger (e.g., `<teddybear>` referring to a specific toy). Although such triggers may appear as a single token to the user, standard tokenizers often split them into multiple subword tokens; for example, `<teddybear>` would be tokenized as two tokens: `teddy` and `bear`. We model this by partitioning the token indices into two sets: $\mathcal{I}_{\text{sks}}$ containing the trigger's sub-token indices (forming a contiguous span $\mathcal{S} = \{i_1, \dots, i_m\}$ where $i_{k+1} = i_k + 1$), and $\mathcal{I}_{\text{other}} = \{1, \dots, T\} \setminus \mathcal{I}_{\text{sks}}$ for all other tokens.

This distinction becomes critical when we examine how different tokens behave under quantization. For each cross-attention block with $L$ layers, $H$ heads, query length $N$, and dimension $d$, we have:

$$\mathbf{Q}^{(\ell,h)} \in \mathbb{R}^{B \times N \times d}, \quad \mathbf{K}^{(\ell,h)}, \mathbf{V}^{(\ell,h)} \in \mathbb{R}^{B \times T \times d}, \quad \mathbf{A}^{(\ell,h)} = \text{softmax}\big(\mathbf{Q}^{(\ell,h)} \mathbf{K}^{(\ell,h)\top} / \sqrt{d}\big) \in \mathbb{R}^{B \times N \times T}.$$

The attention mechanism controls image generation through $\mathbf{A}^{(\ell,h)}\mathbf{V}^{(\ell,h)}$, where keys $\mathbf{K}^{(\ell,h)}$ and values $\mathbf{V}^{(\ell,h)}$ encode text tokens while attention weights $\mathbf{A}^{(\ell,h)}$ determine their spatial influence. Crucially, personalized tokens exhibit unique quantization sensitivity across all these components.

**Isolating token-specific quantization effects.** To understand which tokens are vulnerable to quantization, we designed a surgical experiment. Rather than quantizing the entire model, we selectively quantize only the K/V rows corresponding to specific tokens while keeping everything else at full precision.

Specifically, for a token span $\mathcal{S} \subseteq \{1, \dots, T\}$ and bit-width $b \in \{8, 4\}$, we construct modified parameters $\Theta^{(\mathcal{S}, b)}$ by applying:

$$\forall(\ell, h) \, \forall i \in \mathcal{S}: \quad \mathbf{K}^{(\ell,h)}_{:,i,:} \leftarrow \mathcal{Q}_b\big(\mathbf{K}^{(\ell,h)}_{:,i,:}\big), \qquad \mathbf{V}^{(\ell,h)}_{:,i,:} \leftarrow \mathcal{Q}_b\big(\mathbf{V}^{(\ell,h)}_{:,i,:}\big), \tag{1}$$

where $\mathcal{Q}_b(\cdot)$ denotes uniform affine quantization with per-row scaling. All other components—including $\mathbf{Q}$, non-$\mathcal{S}$ rows of $\mathbf{K}/\mathbf{V}$, and other network modules—remain at FP16. This targeted approach lets us measure each token's individual contribution to quantization error.

**Measuring Quantization Errors.** We measure how much the image changes when only a token's K/V rows are quantized. Given a dissimilarity metric $\mathcal{L}$ with "smaller is better," the per-token sensitivity is

$$\Delta_i(b) = \mathcal{L}\big(\mathbf{y}^{(i,b)}, \mathbf{y}^\star\big), \qquad b \in \{8, 4\}. \tag{2}$$

We use two metrics to quantify this change: (1) *Mean-Squared-Error (MSE)* tracks visible degradations such as blur, noise, and artifacts; (2) *CLIP score (Radford et al., 2021)* that used as a complementary check for semantic drift.

We aggregate sensitivities over token groups and spans to compare triggers vs. other tokens:

$$\Delta_{\text{sks}}(b) = \frac{1}{|\mathcal{I}_{\text{sks}}|} \sum_{i \in \mathcal{I}_{\text{sks}}} \Delta_i(b), \qquad \Delta_{\text{other}}(b) = \frac{1}{|\mathcal{I}_{\text{other}}|} \sum_{i \in \mathcal{I}_{\text{other}}} \Delta_i(b). \tag{3}$$

For *multi-subtoken triggers* ($m > 1$), we additionally visualize their span-level sensitivity (see Appendix Fig. 5).

Table 1: **Quantitative Comparison.** Results of different quantization methods for personalized diffusion checkpoints.

| Selection Method | Quantization Method | Bit-width (W/A) | Param Storage (× smaller) | Bit-Op (× fewer) | MS-COCO FID(↓) | MS-COCO CLIP(↑) | PartiPrompts FID(↓) | PartiPrompts CLIP(↑) |
|---|---|---|---|---|---|---|---|---|
| | Full Precision | 32/32 | 1× | 1× | 10.96 | 0.315 | 9.77 | 0.336 |
| | PTQD | 8/8 | 4× | 16× | 32.78 | 0.254 | 32.25 | 0.251 |
| | | 4/8 | 8× | 32× | 34.36 | 0.243 | 37.61 | 0.242 |
| | | 8/4 | 4× | 32× | 242.11 | 0.072 | 244.84 | 0.077 |
| | | 4/4 | 8× | 64× | 253.52 | 0.068 | 247.66 | 0.074 |
| | Q-Diffusion | 8/8 | 4× | 16× | 27.16 | 0.261 | 23.44 | 0.267 |
| | | 4/8 | 8× | 32× | 30.82 | 0.255 | 30.97 | 0.260 |
| | | 8/4 | 4× | 32× | 223.11 | 0.082 | 218.93 | 0.084 |
| | | 4/4 | 8× | 64× | 231.83 | 0.073 | 232.19 | 0.075 |
| *Check-in* | TFMQ-DM | 8/8 | 4× | 16× | 24.34 | 0.279 | 21.66 | 0.299 |
| | | 4/8 | 8× | 32× | 31.69 | 0.266 | 29.47 | 0.279 |
| | | 8/4 | 4× | 32× | 167.22 | 0.138 | 173.51 | 0.141 |
| | | 4/4 | 8× | 64× | 192.48 | 0.122 | 189.39 | 0.119 |
| | DGQ | 8/8 | 4× | 16× | 15.24 | 0.291 | 13.26 | 0.317 |
| | | 4/8 | 8× | 32× | 22.45 | 0.283 | 20.43 | 0.287 |
| | | 8/4 | 4× | 32× | 53.39 | 0.248 | 55.13 | 0.244 |
| | | 4/4 | 8× | 64× | 64.11 | 0.241 | 67.42 | 0.247 |
| | ***TAQ*** | 8/8 | 4× | 16× | 11.03 | 0.297 | 10.49 | 0.327 |
| | | 4/8 | 8× | 32× | 13.74 | 0.292 | 12.37 | 0.310 |
| | | 8/4 | 4× | 32× | 38.84 | 0.264 | 36.91 | 0.265 |
| | | 4/4 | 8× | 64× | 47.73 | 0.257 | 45.38 | 0.259 |

**Observation: Personalized tokens are vulnerable.** Personalized triggers (e.g., `<sks>`) occupy a distinct *text-side* regime: their cross-attention weights concentrate sharply on a few K/V rows; their K/V entries are heavy-tailed; and small changes to those rows yield large output changes (large $\Delta_i(b)$). This directly implicates the token-specific K/V rows as *quantization-vulnerable* and motivates treating them differently during quantization.

**Trigger-Aware Quantization (TAQ).** Based on these insights, TAQ adopts a selective quantization strategy: protect vulnerable personalized tokens while aggressively quantizing the rest.

We introduce binary masks to identify protected token positions. For keys and values, $\mathbf{M}_{KV} \in \{0,1\}^{H \times T \times 1}$ is broadcast across batch, layers, and channels. For attention scores, $\mathbf{M}_A \in \{0,1\}^{B \times N \times T}$ protects the same positions. Both masks equal 1 at trigger token indices: $\mathbf{M}_{[\cdot]}[...,i,...] = 1$ if and only if $i \in \mathcal{I}_{sks}$.

The quantization selectively applies $a$-bit quantization to non-trigger elements:

$$\tilde{\mathbf{K}} = \mathbf{M}_{KV} \odot \mathbf{K} + (1 - \mathbf{M}_{KV}) \odot \mathcal{Q}_a(\mathbf{K}), \quad \tilde{\mathbf{V}} = \mathbf{M}_{KV} \odot \mathbf{V} + (1 - \mathbf{M}_{KV}) \odot \mathcal{Q}_a(\mathbf{V}), \quad \tilde{\mathbf{Q}} = \mathcal{Q}_a(\mathbf{Q}) \quad (4)$$

$$\hat{\mathbf{A}}^{(\ell,h)} = \mathbf{M}_A \odot \mathbf{A}^{(\ell,h)} + (1 - \mathbf{M}_A) \odot \mathcal{Q}_a(\mathbf{A}^{(\ell,h)}) \quad (5)$$

Here, $\mathcal{Q}_a(\cdot)$ applies $a$-bit quantizer (uniform quantization or logarithmic quantization). For multi-token triggers spanning indices $\mathcal{S}$, all masks protect the entire span: $\mathbf{M}_{[\cdot]}[...,i,...] = 1$ for all $i \in \mathcal{S}$.

# 4 EXPERIMENTS

## 4.1 EXPERIMENTAL SETUP

**Personalized Checkpoints Repository and Repo-Prompts Dataset** We introduce a repository of 1,000 personalized text-to-image checkpoints fine-tuned on Stable Diffusion 1.5. The collection spans 20 concept categories with 50 temporal versions each. Each checkpoint includes metadata for retrieval. To evaluate personalized checkpoint selection from user descriptions (e.g., "the April-created bear model"), we present REPO-PROMPTS, a dataset of 500 natural language queries comprising standard selection cases, ambiguous queries requiring clarification, and no-match cases. Systems are evaluated on correct matching, ambiguity recognition, and appropriate rejection of invalid queries (see Appendix A for details).

**Weights: 8-bit, Activations: 8-bit**     **Weights: 8-bit, Activations: 4-bit**

Full Precision     DGQ     **Ours**         Full Precision     DGQ     **Ours**

Checkpoint <person-v5> :
"a photo of <person-v5> on a black and white mountain ridge."

Checkpoint <painting-v5> :
"Boat on a lake, with misty mountains in <painting-v5> style."

Checkpoint <cat-v5> :
"a stone path in the foreground, <cat-v5>, with green foliage."

Checkpoint <drawing-v5>:
"A sunny park with <drawing-v5> style."

Figure 4: **Qualitative Comparison.** More qualitative results can be seen in Figure 6 and 7

**Baselines.** We evaluate four fixed-precision PTQ methods—Q-Diffusion (Li et al., 2023a), PTQD (He et al., 2023), TFMQ-DM (Huang et al., 2024), and DGQ (Ryu et al., 2025)—none of which performs intent-aligned checkpoint selection; all methods (and ours) use the same personalized checkpoints and prompts to isolate quantization effects. For selection, we include Random (uniform sampling within the prompt's category), Reranker (cross-encoder re-ranking by prompt–description semantic similarity, we use Qwen3-Reranker-4B as implementation (Zhang et al., 2025)), and Stylus (Luo et al., 2024) (cosine-similarity retrieval over checkpoint-description and prompt embeddings).

Table 2: **Checkpoint selection results.** Intent Score (1–5 Likert scale) represents the average LLM judge rating across evaluation dimensions. Intent-Alignment Win Rate shows the percentage of times our method (*Check-in*) was preferred over each baseline in pairwise human evaluations; our method has no win rate against itself (–).

| Selection Method | Intent Score | Intent-Alignment Win Rate (%) |
|---|---|---|
| Random | 2.14 ± 0.82 | 89.1 |
| Reranker | 3.21 ± 0.76 | 85.7 |
| Stylus | 3.68 ± 0.69 | 82.1 |
| **Check-in (Ours)** | **4.42 ± 0.51** | – |

**Weight quantization.** Since our method focuses on activation quantization, we tested it by applying the same weight quantization to both our method and the baseline. We used two techniques: Adaround (Nagel et al., 2020) and BRECQ (Li et al., 2021). Block reconstruction was applied to residual and transformer blocks. The dataset used for calibration during quantization was generated using 64 captions from the personalized calibration dataset (details in Appendix D).

### 4.2 QUANTITATIVE RESULTS

**Automatic Benchmarks.** We evaluated our model on MS-COCO and PartiPrompts datasets using CLIP (Radford et al., 2021) and FID (Seitzer, 2020) scores. Since we use personalized checkpoints for categories (e.g., cat and person), we adapted the evaluation prompts accordingly: 30% of prompts that matched our personalized categories remained unchanged, while 70% were augmented by appending our style checkpoint to ensure compatibility. The results are presented in Table 1. TAQ significantly outperforms all baseline quantization methods across both datasets. On MS-COCO, TAQ achieves FID scores of 11.03 (W8A8) and 13.74 (W4A8), closely approaching the full-precision baseline (10.96) while reducing parameter storage by 4-8× and bit-operations by 16-32×. The CLIP scores remain robust at 0.297 and 0.292, with minimal degradation from full precision (0.315). Notably, while baseline methods suffer catastrophic failure with 4-bit activations (FID more than 160), TAQ maintains relative stability with FID scores of

38.84 (W8A4) and 47.73 (W4A4), demonstrating superior resilience under aggressive quantization. The qualitative results can be seen in Figure 4.

**LLM as a Judge.** We use an LLM judge to evaluate checkpoint-user request alignment across four dimensions: subject fit, style match, temporal fit, and context appropriateness (1-5 Likert scale each (Gu et al., 2024)). Our metric, *Intent Score*, averages scores across these dimensions. To mitigate positional bias (Zheng et al., 2023), we evaluate with flipped option orders and only count consistent results. Using the same experimental setup as human evaluations, Table 2 shows our *Check-in* method achieves the highest Intent Score of 4.42. Full prompts are in appendix Table 10.

**Human Evaluations.** We evaluate our checkpoint selection method against three baselines (Random, Reranker, Stylus) using 500 prompts from REPO-PROMPTS, MS-COCO, and PariPrompts datasets. Three human raters conduct 1,500 pairwise comparisons (500 prompts × 3 baselines' comparision), selecting which checkpoint better aligns with the text prompt. Our metric, *Intent-Alignment Win Rate*, measures the percentage of times our method (*Check-in*) was preferred. Results in Table 2 show win rates of 89.1% (Random), 85.7% (Reranker), and 82.1% (Stylus), averaging 85.6 across baselines.

### 4.3 ABLATION STUDY

**Effect of *Check-in*.** Table 3 shows the clarification serves as the primary driver of intent alignment improvement, elevating the score from a baseline of 3.21 to 4.18. Metadata reasoning alone provides moderate benefits (3.98), suggesting it captures complementary selection criteria. When combined, they achieve the highest intent score of 4.42 with CLIP of 0.297. The modest FID reduction from 11.74 to 11.03 aligns with expectations, as *Check-in* optimizes for selection intent score rather than generation quality.

**Effect of trigger-aware quantization.** Table 4 shows at W8/A8 bits with linear quantization, separation reduces FID from 15.83 to 11.04. The logarithmic quantizer performs poorly without separation, achieving FID of 17.62, but becomes competitive with separation at 13.67. This pattern intensifies at 8/4 bits: logarithmic quantization with separation achieves the best FID of 38.22, outperforming linear quantization at 44.53. These results confirm that trigger tokens are crutial for post-training quantization.

**Effect of each component.** Table 5 demonstrates that *Check-in* and TAQ are complementary mechanisms. *Check-in* improves intent alignment from 3.22 to 4.42 by selecting more appropriate checkpoints, while TAQ reduces FID from 15.22 to 11.03 at 8/8 bits through better quantization. Their combined effect becomes critical at lower precision: at 8/4 bits, using both methods achieves FID of 38.22 and CLIP of 0.265, whereas TAQ alone yields FID of 39.53 with CLIP of 0.261.

Table 3: Effect of *Check-in* components

| Clarify | Metadata | Intent↑ | FID↓ | CLIP↑ |
|---|---|---|---|---|
| ✗ | ✗ | 3.21 | 11.74 | 0.289 |
| ✗ | ✓ | 3.98 | 11.61 | 0.291 |
| ✓ | ✗ | 4.18 | 11.23 | 0.293 |
| ✓ | ✓ | **4.42** | **11.03** | **0.297** |

Table 4: Effect of TAQ and trigger tokens separation

| Bits(W/A) | Quantizer | Separate trigger tokens | FID↓ | CLIP↑ |
|---|---|---|---|---|
| 8/8 | Linear | ✗ | 15.83 | 0.292 |
| 8/8 | Linear | ✓ | **11.04** | **0.298** |
| 8/8 | Logarithmic | ✗ | 17.62 | 0.287 |
| 8/8 | Logarithmic | ✓ | 13.67 | 0.294 |
| 8/4 | Linear | ✗ | 54.12 | 0.245 |
| 8/4 | Linear | ✓ | 44.53 | 0.262 |
| 8/4 | Logarithmic | ✗ | 56.21 | 0.249 |
| 8/4 | Logarithmic | ✓ | **38.22** | **0.265** |

Table 5: Effect of component synergy

| Bits(W/A) | *Check-in* | TAQ | Intent↑ | FID↓ / CLIP↑ |
|---|---|---|---|---|
| 8/8 | ✗ | ✗ | 3.22 | 15.22 / 0.286 |
| | ✓ | ✗ | 4.39 | 14.81 / 0.294 |
| | ✗ | ✓ | 3.21 | 11.74 / 0.289 |
| | ✓ | ✓ | 4.42 | 11.03 / 0.297 |
| 8/4 | ✗ | ✗ | 3.21 | 54.12 / 0.245 |
| | ✓ | ✗ | 4.40 | 51.31 / 0.253 |
| | ✗ | ✓ | 3.23 | 39.53 / 0.261 |
| | ✓ | ✓ | **4.41** | **38.22 / 0.265** |

## 5 CONCLUSION

We introduced PersonalQ, a unified system for selecting and serving personalized diffusion models under tight memory budgets. Our *Check-in* reasons over repository metadata and prompt semantics, maps generic nouns to trigger tokens, and resolves ambiguity, outperforming retrieval baselines on REPO-PROMPTS and in human/LLM evaluations. Complementing this, *TAQ* preserves trigger-token pathways while quantizing non-personalized components, delivering 4–8× parameter compression and up to 75% serving-time GPU-memory reduction with minimal quality loss (e.g., W8A8 FID 11.03 vs. 10.96 FP32). Together, these components enable intent-aligned selection and scalable, fidelity-preserving inference across large personalized checkpoint repositories.

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

## A  PERSONALIZED CHECKPOINTS REPOSITORY AND REPO-PROMPTS DATASET

**Personalized Checkpoints Repository** Our checkpoint collection comprises 20 concept categories (`<dog>`, `<cat>`, `<car>`, `<house>`, `<shoe>`, `<flower>`, `<painting>`, `<person>` ...), each with 50 temporal versions (`v1–v50`), totaling 1000 checkpoints. Each checkpoint is fine-tuned from Stable Diffusion 1.5 using 3–5 concept images over 800 training steps with AdamW optimization (learning rate $\eta = 1e$-6, momentum parameters $\beta_{1,2} = (0.9, 0.999)$). The resulting models are stored as safetensors files under Git LFS, accompanied by YAML metadata containing timestamps, SHA-256 checkpoint identifiers, and predefined style tags.

**Repo-Prompts Dataset** To assess checkpoint selection capabilities, we introduce REPO-PROMPTS, a dataset of 500 natural language queries targeting specific personalized models. Users typically request checkpoints through informal

Table 6: Complete list of 20 concept categories with 50 versions each

| Personalized Concepts | | | |
|---|---|---|---|
| <dog> | <cat> | <person> | <bear> |
| <horse> | <car> | <toy> | <watch> |
| <bag> | <chair> | <house> | <building> |
| <bridge> | <flower> | <tree> | <mountain> |
| <painting> | <drawing> | <logo> | <toy> |

descriptions ("the April dog model", "latest version", "anime style from last week"), requiring systems to map temporal references, concept types, and style attributes to correct checkpoints. The dataset comprises 500 evaluation instances: 350 (70%) standard selection cases with clear single matches, 100 (20%) ambiguous queries requiring clarification, and 50 (10%) no-match cases where no valid checkpoint exists.

Table 7: Evaluation categories with representative examples from Repo-Prompts.

| Category | Example Query | Expected Response |
|---|---|---|
| Temporal | "April anime build" | Select: ckpt_anime_apr_v2 |
| Version | "v2 sketch model" | Match exact version |
| Style | "Realistic urban" | Match all style tags |
| Usage | "Yesterday's model" | Select by usage metadata |
| Ambiguous | "Spring sketch" | Clarify: "April or May?" |
| No-match | "December watercolor" | Return: "No matches" |

Example Instance of REPO-PROMPTS:

```
{
  "instance_id": "rp_001",
  "natural_language_query": "Anime night city, the April build",
  "candidate_pool": ["ckpt_anime_apr_v2", "ckpt_anime_may_v3", "..."],
  "ground_truth": {
    "checkpoint_id": "ckpt_anime_apr_v2",
    "requires_clarification": false,
    "no_match": false
  }
}
```

Each checkpoint contains unique identifiers, style tags (e.g., anime, sketch, realistic), and timestamps. Evaluation instances follow the schema: {instance_id, query, candidate_pool, ground_truth: {checkpoint_id|null, requires_clarification, no_match}}.

## B DETAILS OF THE *Check-in* LLM

We provide the complete *Check-in* LLM prompt in Tab. 8. The prompt uses a two-stage Chain-of-Thought (CoT) approach: checkpoint selection then trigger token replacement.

For selection, we feed the top 10 checkpoints (retrieved via embedding similarity) into Gemini 2.5 Flash's context Team et al. (2023). The prompt guides evaluation across temporal constraints, style preferences, and environment compatibility. When multiple checkpoints match, the LLM generates a single clarification question using natural language rather than technical terms. For trigger mapping, Gemini 2.5 flash identifies generic nouns and replaces them with corresponding trigger tokens. Separating these stages improved intent score compared to a unified approach and eliminated trigger token hallucination.

## C ABLATION STUDY OF MLLM BACKBONES

To deploy models on memory-constrained devices, we evaluated all backbones via API to avoid GPU memory requirements on our hardware. We tested three models: the open-source Qwen2.5-VL-72B (Bai et al., 2025) and two closed-source models, GPT-4o (Hurst et al., 2024) and Gemini 2.5 Flash (Team et al., 2023). As shown in Table 9, all

Table 8: **_Check-in_ LLM prompt**. Checkpoint selection and trigger token mapping.

| Available Checkpoints Metadata in Repository |
|---|

`cat-v3`: Photorealistic gray domestic short-hair cats. Trained on beach/coastal scenes with natural lighting. _Trigger:_ `<cat-v3>`, _Created:_ 2025-01-11, _Version:_ 3

`tooncat`: Stylized cartoon cats with vibrant colors and exaggerated features. Best for whimsical illustrations. _Trigger:_ `<tooncat>`, _Created:_ 2024-12-15, _Version:_ 2

`siamese-ckpt`: Siamese cats with blue eyes and color-point patterns. Optimized for indoor settings. _Trigger:_ `<siamese-ckpt>`, _Created:_ 2025-01-20, _Version:_ 4

`tabby-orange`: Orange tabby cats with striped patterns. Excels at outdoor garden/forest scenes. _Trigger:_ `<tabby-orange>`, _Created:_ 2024-11-30, _Version:_ 1

[...additional checkpoints...]

| **User Request:** _"Generate a photo of the latest cat model on a beach at sunset"_ |
|---|

### Task 1: Checkpoint Selection

**Temporal Requirements:**

- Does the request contain time-sensitive words like "latest", "newest", "recent", or "updated"?
- If yes, sort checkpoints by their creation date and prioritize the most recent ones
- For version references ("v2", "version 3"), match against the version field

**Style Requirements:**

- Identify style keywords: "photo"/"photorealistic" → realistic checkpoints, "cartoon"/"anime" → stylized checkpoints
- Match these against each checkpoint's description and style characteristics
- Default to photorealistic if no style is specified but "photo" or "picture" is mentioned

**Environment Matching:**

- Extract location keywords from the request ("beach", "indoor", "forest", "studio")
- Compare with each checkpoint's training environment and optimal settings
- Prioritize checkpoints trained on matching environments for better quality

**When Multiple Checkpoints Match:**

If several checkpoints satisfy the requirements, ask ONE specific clarification question that highlights the key difference between options. For example: _"Would you prefer the photorealistic gray cat (cat-v3) or the Siamese cat with blue eyes (siamese-ckpt)?"_

### Task 2: Trigger Token Replacement

**Identification Process:**

- Find all generic nouns in the prompt that match the checkpoint's subject ("cat", "kitten", "feline")
- Look up the checkpoint's specific trigger token from its metadata
- Note any aliases or alternative triggers that might work

**Replacement Rules:**

- Replace the generic noun with the exact trigger token, preserving grammar
- Keep all other words unchanged — only modify the subject references
- If multiple instances exist, replace all of them consistently
- Never replace proper nouns or unrelated terms

**Expected Output Format:**

`Selected Checkpoint:` cat-v3 (newest checkpoint, matches "latest" requirement and beach setting)
`Replacements Made:` "cat" → `<cat-v3>`
`Final Prompt:` "Generate a photo of the latest cat-v3 model on a beach at sunset"

three models achieved comparable performance. Notably, Gemini 2.5 Flash demonstrated the fastest inference speed while maintaining competitive intent recognition scores, offering an optimal balance between speed and intent score for our use case.

Table 9: **MLLM ablation on Repo-Prompts.** We report Intent Score along with a full inference-time breakdown.

| MLLM backbone | Performance (Repo-Prompts) Intent Score | Inference time (s) | | | | | Multi-turn |
|---|---|---|---|---|---|---|---|
| | | Retrieve | Reason | Clarify | Generation (W8A8) | End-to-End | |
| GPT-4o (API) | 4.42 | 1.3 | 20.31 | 11.17 | 12.31 | 45.09 | 2.1 |
| Gemini 2.5 Flash (API) | 4.38 | 1.3 | 16.54 | 8.28 | 12.31 | 38.43 | 2.3 |
| Qwen2.5-VL-72B (API) | 4.35 | 1.3 | 25.89 | 10.35 | 12.31 | 49.85 | 2.5 |

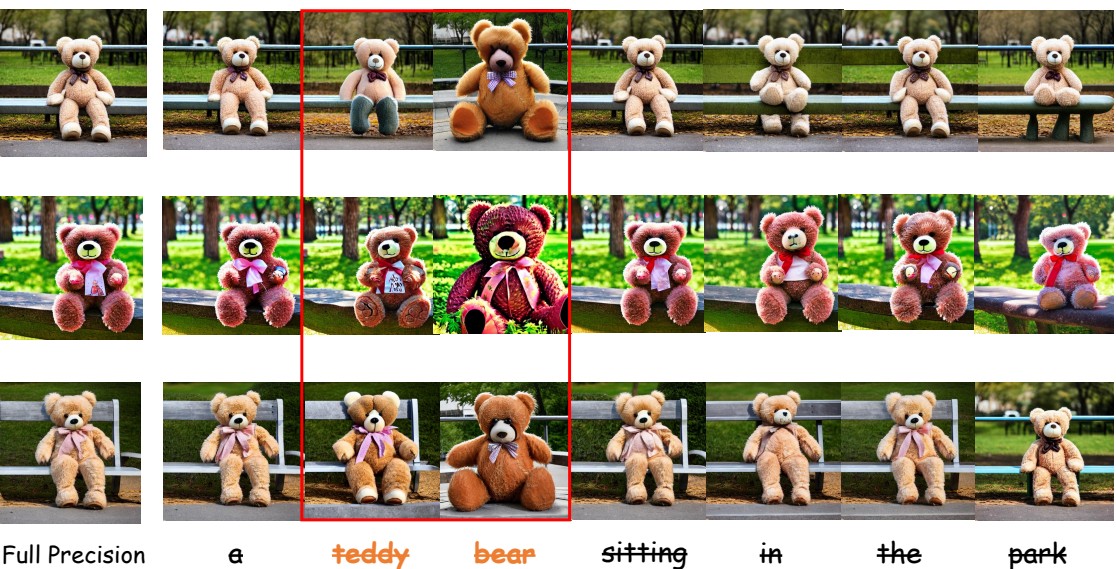

Full Precision    a    teddy    bear    sitting    in    the    park

(a) Token-wise sensitivity. Only one token is quantized at a time; all others remain full precision. The trigger `<teddybear>` splits into two subtokens, `teddy` and `bear` (red box); quantizing either subtoken causes the largest visual degradation. Results shown across three independently trained personalized checkpoints that share the same trigger.

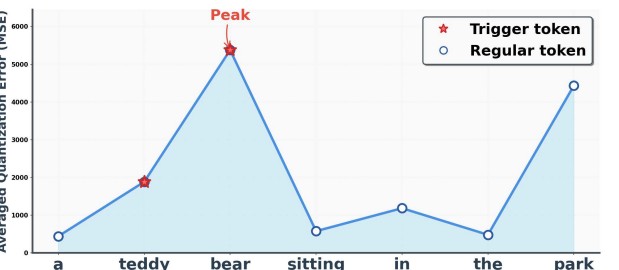

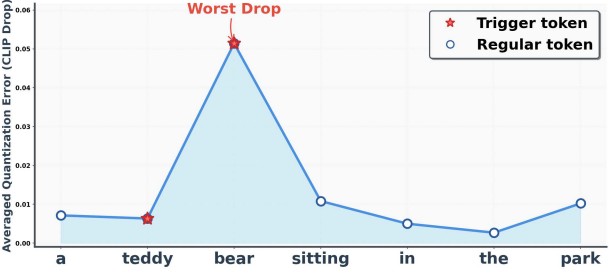

(b) Averaged MSE when quantizing each token (higher is worse). Error peaks at the trigger token.

(c) Averaged CLIP-score drop when quantizing each token (higher is worse). The trigger token yields the largest drop.

Figure 5: **Trigger span (e.g., `<teddybear>` → `teddy, bear`) is vulnerable under quantization across checkpoints.** We evaluate three personalized checkpoints that all share the same trigger `<teddybear>`, measuring per-token sensitivity to cross-attention Key/Value-row quantization by quantizing one token at a time while keeping all others in full precision. Visual results and aggregate metrics (averaged over the three checkpoints) show that the trigger subtokens are far more fragile than common words under 4-bit quantization. *Prompt used:* "a `<teddybear>` sitting in the park", where the trigger span is {`teddy,bear`}.

## D   CALIBRATION PROMPTS AND PROCEDURE

We construct a compact yet diverse prompt corpus by templating the learned trigger token into varied contexts. Each prompt follows:

Table 10: **LLM as a judge.** Evaluation prompt for intent matching between user prompts and checkpoint selection.

| Intent Matching Evaluation | |
|---|---|
| **Input:** | *User Prompt:* "{user_prompt}" |
| | *Selected Checkpoint:* {checkpoint_name} |
| | *Metadata:* {checkpoint_description} |
| | *Alternative Checkpoints:* {alternatives} |
| **Scoring Criteria (1–5 scale)** | |
| **Subject Match:** | **5**: Exact subject(s) match ("cat" → cat model) |
| | **3**: Reasonable interpretation ("pet" → cat specialist) |
| | **1**: Clear mismatch ("landscape" → portrait model) |
| **Style Intent:** | **5**: Perfect style alignment with user request |
| | **3**: Plausible style inference from context |
| | **1**: Style clearly contradicts intent |
| **Temporal Match:** | **5**: Meets temporal requirements (e.g., "latest" → newest) |
| | **3**: Reasonable temporal interpretation |
| | **1**: Temporal constraints ignored |
| **Context Fit:** | **5**: Perfect fit for scenario/use case |
| | **3**: Generally appropriate for context |
| | **1**: Contextually inappropriate |
| **Scoring: Comparison:** | Intent Score = $\frac{1}{4}\sum$(subject, style, temporal, context) |
| | Forced choice: "Selected Better" *or* "Alternative [X] Better" |
| **Output Format:** | {``subject'': X, ``style'': X, ``temporal'': X, ``context'': X, |
| | ``intent_score'': X.X, ``best_choice'': ``...'', ``reasoning'': ``..."} |

Table 11: **Calibration prompt synonym pools.** One value is sampled per category.

| Category | Count | Values |
|---|---|---|
| Scenes | 16 | on a sunny beach; in a lush green park; inside a cosy living-room; on a snowy mountain at sunset; running across a golden wheat field; under cherry-blossom trees; posing in front of the Eiffel Tower; sitting on a skateboard at a skatepark; playing fetch by a forest lake; splashing in a backyard pool; walking through bustling city streets; lying on a vintage Persian rug; climbing a rocky cliff by the sea; resting beside a campfire; sitting in a canoe on calm water; wrapped in a warm blanket during a snowfall. |
| Time / lighting | 8 | at dawn; at golden hour; at blue hour; on an overcast day; under neon lights at night; at high noon; during a thunderstorm; at twilight. |
| Styles | 13 | high-resolution photograph; analog film style; soft focus portrait; 35 mm film; ultra-wide angle shot; macro shot; HDR; vintage Polaroid; cinematic still; professional studio lighting; low-key lighting; minimalist composition; aerial drone view. |
| Optional actions | 12 | wearing sunglasses; jumping over an obstacle; catching a frisbee; tilting its head; sleeping peacefully; wagging its tail; smiling at the camera; with its tongue out; howling playfully; sniffing a flower; splashing water; wrapped in a scarf. |

```
a photo of a <sks> {class}, {scene}, {time}, {style}, {action}
```

where `<sks>` is the trigger token and {`class`} is the target category (e.g., *dog*). At instantiation time, we sample exactly one element from each category listed in Table 11. This design anchors subject identity while varying background, time/lighting, photographic style, and an optional pose/action to elicit a broad span of activations during calibration.

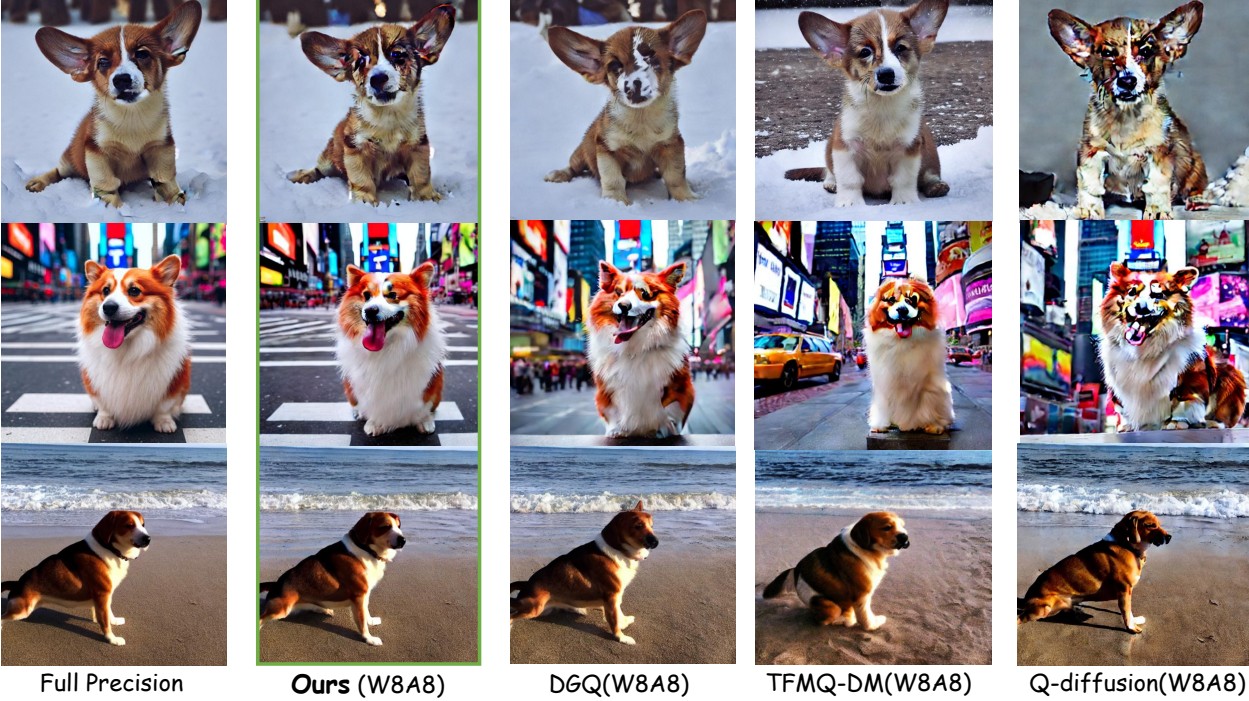

(a) Columns show different personalized `<drawing>` checkpoints; rows use the same prompts triggered by `<drawing1>`, `<drawing2>`, and `<drawing3>` to isolate checkpoint effects: (1) "A `<drawing1>`-style photo of a city rooftop building."; (2) "A `<drawing2>`-style photo of the Eiffel Tower."; (3) "A `<drawing3>`-style photo of a cartoon drawing."

(b) Columns show different personalized `<dog>` checkpoints (e.g., `<dog1>`, `<dog2>`, ...), while rows use the same prompts to isolate the effect of checkpoint versioning: (1) "A photo of `<dog1>` in the snow, with a blurry forest background."; (2) "A photo of `<dog2>` at the Times Square pedestrian intersection."; (3) "A photo of `<dog3>` on the beach with waves in the background."

Figure 6: **More qualitative results for `<drawing>` and `<dog>` categories under 8-bit weights.** Subfigure (a) shows `<drawing>` and subfigure (b) shows `<dog>`.

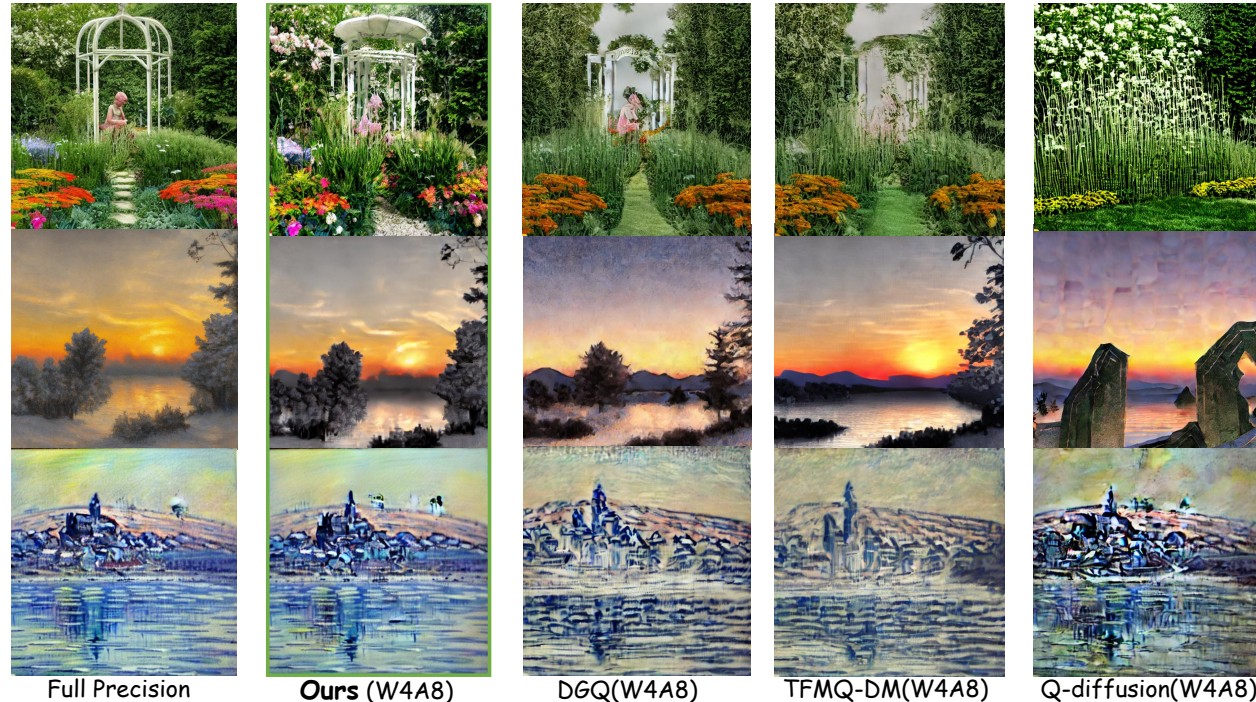

(a) Columns show different personalized `<painting>` checkpoints, while rows use the same prompts triggered by `<painting1>`, `<painting2>`, and `<painting3>` to isolate the effect of checkpoint versioning: (1) "A `<painting1>`-style photo of a flower garden with a white wrought-iron gazebo, stone path, and colorful blooms."; (2) "A `<painting2>`-style photo of a tranquil lakeside at sunset with low mountains, tree silhouettes, and calm-water reflections."; (3) "A `<painting3>`-style photo of a blue-toned cartoon painting of a coastal hillside village with a church tower and bay reflections."

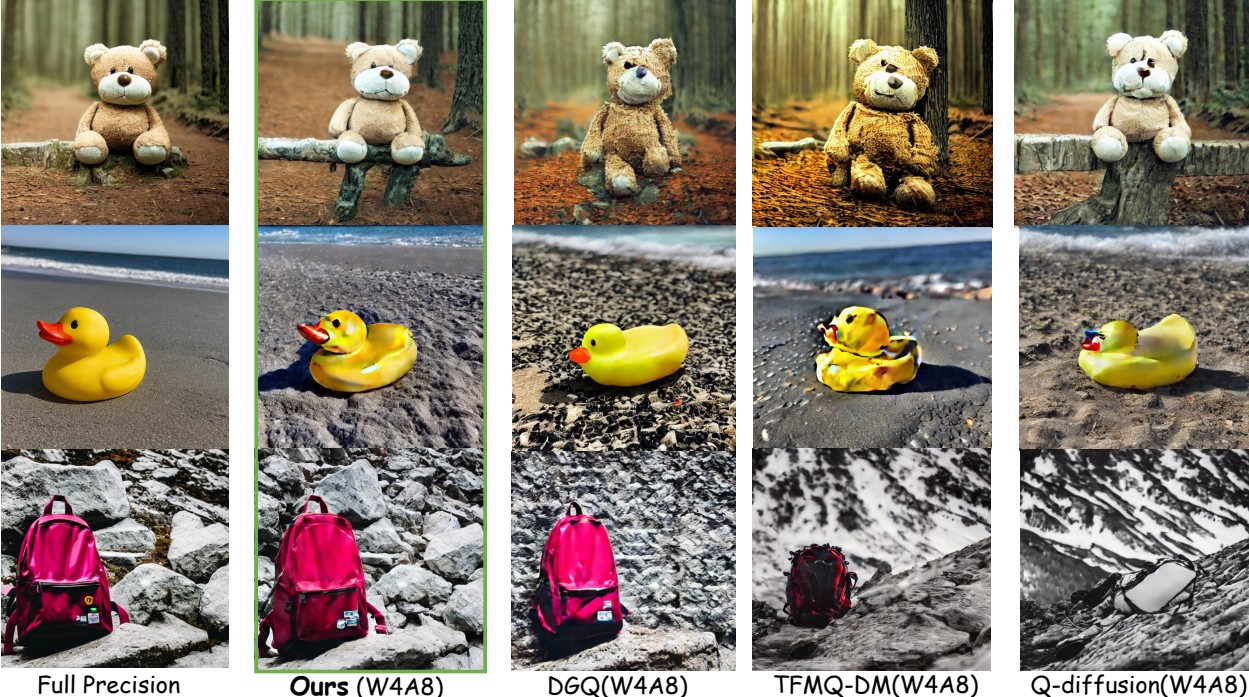

(b) Columns show different personalized `<toy>` checkpoints, while rows use the same prompts to isolate the effect of checkpoint versioning: (1) "A photo of `<toy1>` sitting on a wooden log in a conifer forest."; (2) "A photo of `<toy2>` on wet sand beside the ocean."; (3) "A photo of `<toy3>` resting on gray rocks in a mountain landscape."

Figure 7: **More qualitative results for `<painting>` and `<toy>` categories under 4-bit weights.** Subfigure (a) shows `<painting>` and subfigure (b) shows `<toy>`.

