# OpenReview forum: "PersonalQ: Select, Quantize, and Serve Personalized Diffusion Models for Efficient Inference"
_ICLR.cc/2026/Conference — ICLR 2026 Conference Withdrawn Submission_

### Official Review · Reviewer_Kcte · 2025-10-17

**Soundness:** 2
**Presentation:** 3
**Contribution:** 1
**Rating:** 2
**Confidence:** 4

**Summary:**

This paper presents PersonalQ, a two-stage system for personalized diffusion model serving. Check-in selects the intended personalized checkpoint via metadata reasoning and LLM-based prompt clarification, while Trigger-Aware Quantization (TAQ) preserves trigger-token features during quantization to maintain generation quality. Experiments on 1,000 checkpoints show improved selection accuracy and memory reduction.

**Strengths:**

1. The writing is clear and well organized, with figures that effectively convey the pipeline and make the method easy to follow.
2. The methodology is presented in a structured and understandable way.

**Weaknesses:**

1. The paper combines two loosely related parts—checkpoint selection (Check-in) and quantization (TAQ)—without providing a necessary algorithmic or conceptual connection between them. The first focuses on metadata-based retrieval and prompt reasoning, while the second focuses on bit-width compression; there is no shared signal, dependency, or experiment that couples the two. As a result, the work feels incoherent and reads as a forced stitching of two independent topics rather than a unified framework.

2. The “Check-in” module lacks genuine originality in its use of LLM-based filtering. Integrating an LLM to refine metadata-based retrieval is conceptually straightforward and offers no personalization-specific innovation—the same mechanism could apply to any retrieval task. The paper should highlight what is uniquely designed for personalized model serving rather than relying on generic LLM reasoning over metadata.

3. The quantization method (TAQ) is overly simplistic—essentially skipping quantization for trigger-token–related key/value activations—and lacks deeper analysis or meaningful insights. The paper should broaden its investigation by examining which non-trigger tokens are also fragile (e.g., rare class/style tokens or time/location terms), testing prompts without explicit special tokens (since some LoRA models on Civitai, for instance, are trained without or do not expose trigger tokens), and analyzing sensitivity across Q vs. K/V vs. A and per-head/per-layer configurations, or exploring any other possible insights that could reveal broader principles beyond the trivial trigger-token masking rule.

4. The claimed “up to 75% GPU-memory reduction” appears unsupported. The results show only theoretical compression ratios without real-device tests or memory profiling. Since trigger-token pathways remain in full precision, the actual saving would be smaller. The paper needs concrete hardware measurements (peak memory, latency, throughput) to justify the claimed efficiency gains.

**Questions:**

See weakness above.

---

### Official Review · Reviewer_B5Tg · 2025-10-31

**Soundness:** 2
**Presentation:** 2
**Contribution:** 3
**Rating:** 4
**Confidence:** 3

**Summary:**

The authors explore a setup where a system consists of hundreds of LoRA checkpoints obtained through fine-tuning of a diffusion model. A user interacts with this system via natural language prompts, without employing specific trigger words associated with individual LoRAs. Firstly, the ambiguity of selecting the best-fit LoRA is addressed through LLM interaction with LoRA-related metadata and clarification questions that are posed to the user. Furthermore, memory constraints are discussed through a new quantization strategy, TAQ, which omits quantization for trigger-word-related K/V rows. This approach is motivated by the observation that trigger-word related tokens are particularly vulnerable to quantization error.

**Strengths:**

1. The paper addresses a real and practical scenario where user intent must be clarified and the most appropriate LoRAs selected to maximize alignment with user intention as well as image quality.

2. The proposed TAQ strategy appears to be a simple adjustment that enables an effective trade-off between memory constraints and generated image quality.

**Weaknesses:**

1. The novelty of the Check-In mechanism as a multi-stage strategy is not clear and lacks explicit comparison with the related work Stylus (Luo et al., 2024). Specifically, it is unclear what changes make the proposed method superior in quality to previous approaches. In Stylus, the authors also employ a multi-stage strategy, while in lines 405–407 it is stated that only cosine-based similarity was used to retrieve relevant LoRAs and the “Composer” stage was omitted, which in their work was also used to filter out LoRAs.

2. The FID and CLIP metrics are presented only in the method ablation section, not in the baseline comparison (Table 2), where an LLM is used as a judge and Human Evaluation results are reported.

3. Although the problem is highly relevant, based on the benchmark and lines 260–261 in the algorithm description, it appears that PersonalQ selects the top-1 candidate LoRA, ignoring scenarios where multiple LoRAs need to be combined (e.g., to merge subject and style). This aspect was previously explored in Stylus, making the comparison less clear and the setup more limited.

**Questions:**

1. Regarding quantization: as I understand, other methods in comparison have been used out of the box. Is the proposed approach compatible with any of them? Similarly, have you experimented with baselines beyond simply disabling quantization—such as trigger-aware calibration targeting these specific K/V rows?

2. In Table 3, for TAQ, it is unclear what drawbacks arise in terms of memory usage and latency, since the Param Storage column lacks sufficient detail. In practice, how do these parameters compare to the fully quantized baseline?

---

### Official Review · Reviewer_cBSy · 2025-11-01

**Soundness:** 3
**Presentation:** 1
**Contribution:** 3
**Rating:** 2
**Confidence:** 4

**Summary:**

This paper addresses the important and practical problem of how to use the large, community-driven repositories of personalized generative models according to user intent. The authors identify that personalized models are highly sensitive to quantization, particularly their "trigger tokens" (which invoke specific objects or styles), and that naive quantization degrades quality.
To overcome this, they propose TAQ (Trigger-Aware Quantization). Concurrently, they propose "Check-in," a retrieval and selection framework to find desired checkpoints from large repositories based on user queries, and introduce the "Repo-Prompt" benchmark to evaluate such retrieval methods. The authors report that TAQ achieves quality close to full precision despite weight reduction, and that "Check-in" achieves an 89% win rate in human preference studies.

**Strengths:**

- Practical Problem: The paper tackles a highly relevant, real-world challenge. The "Check-in" methodology for handling massive repositories and the analysis of trigger token sensitivity are both practical and insightful.

- New Benchmark: The proposal of the "Repo-Prompt" benchmark is a valuable contribution to the community, as it provides a standardized way to evaluate retrieval methods in this domain.

- Intuitive Method: The core idea of TAQ—preserving precision for critical trigger words and their corresponding input/output computations—is a very intuitive and straightforward approach to minimizing fidelity loss.

**Weaknesses:**

Despite the promising contribution, this paper suffers from a critical, overriding flaw in its presentation, along with several other significant technical weaknesses.

1. Fundamental Weakness: Critical Violation of Formatting Guidelines. The primary reason for the 'Presentation: 1 (Poor)' score and 'Reject' rating is a clear and severe violation of ICLR 2026 formatting policy. The submission appears to have manipulated page margins and/or horizontal spacing to fit significantly more content into the 9-page limit than is permissible. This represents a significant unfair advantage over all other submissions that adhered to the guidelines. This violation alone is sufficient grounds for rejection, as it fundamentally undermines the fairness of the review process.

2. Unconvincing Qualitative Results: The qualitative results in the Appendix (Figs 6, 7) are not compelling. All methods, including the baseline and "Ours," appear to produce low-quality, degraded outputs. If these are the best-case examples, it calls into question whether the proposed quantization method offers any practical visual benefit.

3. Missing Key Baseline: The quantization experiments are missing a critical baseline comparison. The paper does not compare TAQ to NF4 (Normal Float) [1], which is a state-of-the-art method for preserving precision in quantized models and is highly relevant to this work.

4. Artificial Experimental Setting: The "Check-in" method relies on checkpoint metadata (e.g., creation date, version) that is unlikely to be known or used by a typical user browsing a large, "in-the-wild" repository. This suggests a significant disconnect between the proposed benchmark and a realistic user scenario.

5. Limited Model Scope: Nearly all experiments are conducted on Stable Diffusion 1.5. There is no validation on other models (e.g., FLUX, SDXL) to demonstrate the generalizability of these findings.

6. Missing LLM Usage Disclosure: The paper does not appear to include an LLM Usage statement, which is mandated by the ICLR 2026 Author Guide, especially if LLMs were used for processing metadata or generating prompts.

7. Minor Presentation Issues:
  - Some sentences appear to be duplicated on Page 6 (Lines 270-273).
  - Typo: An opening quotation mark is incorrectly formatted as a closing quote (L36).
  - The 'trigger words' for MS COCO are not clearly defined.
  - The References section runs directly into the Appendix without a \clearpage.

[1] Dettmers, Tim, et al. "Qlora: Efficient finetuning of quantized llms." Advances in neural information processing systems 36 (2023): 10088-10115.

**Questions:**

1. (See Weaknesses) The formatting violation (Weakness #1) is the most critical issue.

2. Missing Rationale (Timestamp): What is the rationale for using timestamp as a feature for model selection? How does it provide more information than simple version control?

3. Missing Rationale (Metadata): Why were other, potentially more semantic, metadata fields not considered for retrieval?

4. Schema-Free Retrieval: Given the issues with metadata (Weakness #4), what are the authors' thoughts on a schema-free retrieval environment, which would seem to be the ideal?

5. Example Request: Could the authors provide a concrete example for the Selection Stage Step 3 where $|\mathbb{C}_F| > 1$?

6. Speed & Memory Benchmarks: The goal of TAQ is to serve multiple models, but what is the actual wall-clock speed and memory usage? The upcast operations required by TAQ likely mean that the ideal speed/memory gains are not realized. Please provide benchmarks comparing TAQ's real-world latency/memory footprint against sequential processing.

### LLM Disclosure
I have used an LLM to assist with improving the grammar, clarity, and polishing of this review. The content, analysis, and final judgments are entirely my own.

---

### Official Review · Reviewer_nQfr · 2025-11-01

**Soundness:** 3
**Presentation:** 3
**Contribution:** 3
**Rating:** 6
**Confidence:** 3

**Summary:**

This manuscript proposes personalQ, an interesting framework that address the ambiguous user prompt matching and quantization model degradation in personalized text-to-image model deployment. The authors introduce Check-In for checkpoint analysis and Type-aware Quantization (TAQ) for high quality inference. The authors also introduce the Repo-Prompt benchmark, and experiments on the benchmark demonstrate the superiority of the proposed method.

**Strengths:**

* The personalized text-to-image model deployment is an interesting and practical topic. The proposed solution is quite practical. It is appreciated such work fill gaps between personalization model deployment and large amount of model serving.

* The manuscript is well written, and easy to understand. The overall structure of the article is clear, and the author has used many charts and figures to illustrate the content.

* The proposed Repo-Prompt benchmark is appreciated. It will serve as important benchmark for future research in this field.

**Weaknesses:**

1. Some related references are missing, and it is suggested to consider the related work in the manuscript.

* https://arxiv.org/html/2406.18820v1

* https://arxiv.org/html/2504.15298v1

2. In the experiment, the authors only consider the Stable Diffusion 1.5, how is the method's performance on other text-to-images models?

3. The Check-in part relies on large-language models for reasoning, which may introduce latency in high-concurrency scenarios. In addition, the authors define the ``Intent'' metric in the experiments, is this metric plausible?


4. In table 1, it seems all methods suffer from the low bit limitation, the FID is much higher for W4A4 scenario, it is suggested that the author discuss some possible solutions for low bit case.

**Questions:**

See the weakness above

---

### Note · Authors · 2025-11-19

**Comment:**

We sincerely thank the reviewers (nQfr, cBSy, B5Tg, and Kcte) for their time and thoughtful feedback on our submission.

After internal discussion, we have decided to withdraw this submission and substantially revise the work offline. We will carefully incorporate the reviewers’ suggestions into a future version.

Thank you again for your efforts and constructive comments.

**Withdrawal Confirmation:**

I have read and agree with the venue's withdrawal policy on behalf of myself and my co-authors.